# Comparison between Experience-Based and Household-Undernourishment Food Security Indicators: A Cautionary Tale

**DOI:** 10.3390/nu12113307

**Published:** 2020-10-29

**Authors:** Luis A. Sandoval, Carlos E. Carpio, Manuel Garcia

**Affiliations:** 1Department of Agribusiness Management, Zamorano University, Francisco Morazán 11101, Honduras; 2Department of Agricultural and Applied Economics, Texas Tech University, Lubbock, TX 79409, USA; carlos.carpio@ttu.edu (C.E.C.); manuel.garcia@ttu.edu (M.G.)

**Keywords:** ELCSA, undernourishment, food security, Guatemala

## Abstract

Food security is a multi-dimensional concept that requires multiple indicators to measure it correctly; however, single food security indicators are often used individually or interchangeably. The misinterpretation of individual food security indicators can have important implications for policy design and implementation. The general objective of this paper is to show the discrepancies that may arise when using two different food security indicators that operate in the same dimension of the food security concept and yield the same outcome (food security status of the household) in three of the scenarios that they might be used: (1) for measuring the prevalence of food insecurity, (2) for understanding its drivers, and (3) for estimating the potential impact of a policy. The specific objectives of this paper are (1) to measure and compare the prevalence of food insecurity in a country using the Latin America Food Security Scale (ELCSA, by its acronym in Spanish) and the household undernourishment indicator, (2) to compare the factors associated with households’ food security status using the two indicators, and (3) to assess the potential use of the two indicators for ex ante policy analysis. Data for the study comes from the 2011 Survey of Living Standards from Guatemala, which collected all the data for estimating the ELCSA and the household level data required for calculating the household undernourishment indicator. Our results indicate considerable differences in the estimated prevalence of food insecurity at the national and regional levels using the two alternative indicators, with ELCSA resulting in higher estimates. Logistic regression models estimated to assess and identify household food insecurity drivers also found large differences in both the direction and magnitude of factors affecting food insecurity using the alternative food security indicators. Finally, the magnitude of the simulated impact of a cash transfer policy varied depending on the food indicator used.

## 1. Introduction

Food security exists when all people, at all times, have physical and economic access to sufficient, safe, and nutritious food to meet their dietary needs and food preferences for an active and healthy life [1]. Four dimensions can be identified from the food security concept: availability, economic and physical access, utilization, and stability. The concept of food security applies to individuals, households, regions, or nations. In particular, a household can be categorized as food secure when all its members have access to the food they need for an active and healthy life [1]. Thus, food security at the household level is usually measured in the access dimension of the food security concept [2]. Prominent examples of indicators developed for this purpose are the Latin American and Caribbean Food Security Scale (ELCSA) and a household undernourishment (or household food energy deficiency) indicator calculated from Household Expenditure Surveys (HES). Although the two indicators are calculated differently, both use information collected at the household level and can be used to estimate the prevalence of food insecurity in a region or country, making them suitable for national-level policy analyses [3,4]. 

ELCSA is an experience-based scale estimated by asking one adult or the head of the household a series of questions aimed to evaluate the household’s food security status [3]. On the other hand, the household undernourishment indicator is constructed by comparing the energy available (kilocalories) from food obtained from various sources to all household members’ energy requirements, information that is available in HES data [4]. ELCSA and the household undernourishment indicator operate in the same dimension of food security (access), refer to the same subject of study (households), and provide the same output (household food security status). Because of these similitudes, researchers and policymakers may use them interchangeably, resulting in substantial differences in households’ classification as food secure or insecure with further implications regarding their inclusion or exclusion from assistance programs [5]. Therefore, the main objective of this paper is to show the discrepancies between ELCSA and household undernourishment in three of the scenarios that they might be used: (1) for measuring the prevalence of food insecurity, (2) for understanding its drivers, and (3) for estimating the potential impact of a policy. The specific objectives of this paper are (1) to measure and compare the prevalence of food insecurity in a country using the ELCSA and the household undernourishment indicator, (2) to compare the factors associated with households’ food security status using the two indicators, and (3) to assess the potential use of the two indicators for ex ante policy analysis.

Data for the study comes from the 2011 National Survey of Living Standards (ENCOVI) from Guatemala. We focus on these two indicators, given their growing importance to assess the status of food security. ELCSA is gaining popularity in Latin America and the Caribbean and is now included as part of nationally representative HES in several countries [6,7]. The household undernourishment indicator is also being used more frequently as more countries are periodically conducting HES. 

Before discussing the literature review, it is important to note the difference between the household undernourishment indicator used in this study with FAO’s prevalence of undernourishment (PoU) indicator. Whereas the household undernourishment indicator is constructed using household survey data, FAO’s PoU uses country-level aggregated and statistical modeling to estimate the undernourished population’s proportion. However, given its aggregate nature, and in contrast to the household undernourishment indicator, PoU cannot be used to provide estimates or analyze food insecurity among sub-groups of a country’s population. 

The rest of this section is structured as follows. First, we review the existing literature comparing food security indicators, both theoretically and empirically. Then we provide detailed explanations of the ELCSA and household undernourishment backgrounds and estimation procedures. Because this paper’s main objective is to compare both food security indicators and not offer a more in-depth understanding of Guatemala’s food insecurity status during 2011, only a brief description of the country’s situation during that year is provided at the end of the literature review. 

### 1.1. Literature Review 

The literature comparing food security indicators concentrates on discussing theoretical advantages and disadvantages of using different indicators to estimate food insecurity [2,8,9]. Little research has been conducted to empirically evaluate food security assessments’ differences when alternative indicators are implemented [5,10]. Moreover, previous studies have not considered either the implications of using different indicators to evaluate factors associated with food insecurity and/or policy analysis.

Regarding studies comparing household-level food access indicators at the conceptual level, Jones et al. provide a compendium and review of metrics to measure food security, including ten indicators/approaches to measuring food access (including ELCSA and the use of HES). This study provides background on the construction and assessments of each indicator and discusses their advantages and disadvantages. Leroy et al. identified nine indicators of food security access and evaluated the indicators regarding their validity and equivalence. Validity was assessed considering the indicator’s relation to a well-defined theoretical concept, their stability upon repeated measurement (reliability), and their ability to provide an unbiased estimate of the concept being measured (accuracy). Equivalence was evaluated based on the indicator consistency across contexts (e.g., population groups). The study showed that most evaluations of indicators’ accuracy had been carried out comparing the indicator with determinants or consequences of the indicator. Leroy et al. ‘s key recommendation is to evaluate the performance of food access indicators for different purposes (e.g., policy analysis) beyond the estimation of food security prevalence in a given population [2,8].

Concerning studies that empirically evaluate food access indicators’ performance, Maxwell et al. compared an experience-based measure of food security similar to ELCSA, the Household Food Insecurity Access Scale (HFIAS), to dietary diversity, food frequency, consumption behaviors, and self-assessment indicators. Data for the study was obtained from a panel survey of rural households in Ethiopia. Their results showed correlations between 0.46 and 0.85 (absolute values) between HFIAS and the other food security indicators, with HFIAS always resulting in higher estimates of the prevalence of food insecurity (between 15 to 75%) than the other indicators. They concluded that HFIAS tends to give higher estimates of the prevalence of food insecurity because it captures physiological anxiety and preferences, which are not severe manifestations of food insecurity. Moreover, they contend, a single occurrence of a food insecurity experience can move a household towards a more critical food insecurity category. Despite the differences in the categorization of food-insecure households, all the indicators exhibited similar trends across time in the incidence of food insecurity. These authors argue that the observed differences in the percentages of households classified as food insecure using different indicators are due to three main reasons. First, the cut-off points for the classification of households as food secure or insecure are not harmonized across indicators; second, different indicators measure different dimensions (or sub-dimensions) of food insecurity; and third, they measure varying severities of food insecurity—some only capture food deprivation, some capture the full range of food insecurity. An important difference of Maxwell et al. study to previous studies comparing indicators is the evaluation of indicators not only in terms of correlation with other indicators but also in terms of the differences in estimates of the prevalence of food insecurity and the discussion of potential causes of the observed differences [5]. 

We only identified one study analyzing the relationship between ELCSA and household undernourishment [11]. The study, conducted in Colombia, used household undernourishment as the reference food security measure to evaluate the external validity of ELCSA. Data for the study was obtained from a survey of 432 households located in three Colombian cities. The study found that households’ food security classification status obtained using ELCSA was similar to that obtained using the undernourishment indicator. However, the study does not include any discussion regarding potential sources of differences between the indicators. 

This study contributes to the literature by empirically comparing two food access indicators (ELCSA and household undernourishment) using a nationally representative sample of a country’s population. The use of a national representative sample allows us to compare overall food insecurity prevalence at the national level and prevalence across two policy-relevant segments of the population (region of residence and income level), which has not been possible in previous studies. We also assess the use of the two indicators to evaluate factors associated with food insecurity and the ex ante analysis of a policy (i.e., a cash transfer) to reduce food insecurity. Thus, the proposed framework of analysis can help researchers evaluate food security indicators’ relative performance for purposes beyond prevalence estimation. 

We also aimed to provide prevalence estimates of food insecurity in Guatemala and identify factors affecting Guatemalan households’ food security status. With a population of 16 million, Guatemala is the most populated country in the Central American region and its largest economy [12]. Even though Guatemala is classified as a middle-income country, it still faces significant challenges in terms of poverty and food security [13]. In 2014, it was estimated that about 60% of the population lived below the poverty line [12]. In terms of food security, Guatemala is among the Latin American countries with the highest food insecurity levels in the region (PoU of 15.8%), only below Haiti, Bolivia, and Nicaragua [14]. 

### 1.2. The Latin America and Caribbean Food Security Scale (ELCSA)

ELCSA is an experience-based food security scale that operates on the access dimension of the food security concept. Created following the United States Department of Agriculture’s (USDA) Household Food Security Survey Module, the indicator evaluates how households experience food security by assessing their concerns and experiences during the last three months using a questionnaire’s responses. More specifically, the indicator seeks to assess the quantity and quality of the household’s diet, its use of coping strategies such as eating less than usual or skipping meals, and feelings of anxiety about food access [2,3]. Survey questions used to calculate experience-based food security indicators such as ELCSA have basis on a theoretical construct of food insecurity as a process that starts with anxiety about having enough food, followed by adjustments in the quantity and quality of the diet, and finally, a reduction in consumption [15,16]. Measurement of food insecurity from the questionnaires for experience-based scales is based on item response theory (IRT) methods. These methods assume that although food insecurity is a latent unobserved trait, it can be measured using the responses to the survey questions [16,17]. 

Several reasons make ELCSA an attractive food security indicator. First, ELCSA is a relatively inexpensive measure of household food security compared to food security measurements based on food consumption and expenditure surveys. Second, since it is a standardized measure, it can be used for cross-country comparisons. Finally, ELCSA does not only classify households as food secure and insecure but also provides different levels of food insecurity (see Table 1). ELCSA has been validated in several Latin American countries and is currently included in national surveys in Brazil, Bolivia, Colombia, El Salvador, Guatemala, and Mexico. ELCSA’s scientific committee report (FAO, 2012) provides more details about the indicator construction and a summary and references of previous studies evaluating and validating the indicator [3]. 

In Guatemala, ELCSA consists of 15 simple "yes" or "no" questions. The household’s food security status is then estimated depending on the number of affirmative answers to the questionnaire. When households do not have children, only the first eight questions are asked. The remaining seven-question are intended to measure children’s food security status only and therefore are omitted for childless households. There are four possible categories for a household’s food security status: food secure, mildly food insecure, moderately food insecure and severely food insecure (see Table 1). Appendix A includes the ELCSA questionnaire currently used in Guatemala.

### 1.3. Household Undernourishment Indicator 

The household undernourishment indicator is a nutrition-based indicator obtained by comparing the caloric requirements of a household with the calories it has available for consumption from all food sources to determine if the household is energy deficient [4]. Data for this indicator comes from household consumption and expenditure surveys, which are part of nationally representative living standard surveys that are periodically conducted in several countries worldwide [18]. In 2007, researchers at the International Food Policy Research Institute (IFPRI) developed a technical guide to measure food security using household expenditure surveys [4]. Their objective was to reduce the gap between more nuanced and costly undernourishment measurements at the individual and household level that were being carried out in small populations, such as dietary recall diaries and more aggregate methods for large populations, such as FAO’s PoU. Smith and Subandoro show how the information from HES can be used to define several household-level food security indicators, including the household level undernourishment indicator, which we estimate and compare to ELCSA in this study [4]. 

ELCSA and the household undernourishment indicator are similar in terms of their overall aim, which is to gauge the food access dimension of food security at the household level. However, they differ in terms of the reference period they refer to, the type of data used to calculate the indicator, the cost and difficulty in calculating the indicators, and in terms of the sub-dimensions of food security they encompass. Concerning the reference period, the household undernourishment uses food acquisition data from the previous 15 days of the interview, whereas ELCSA questions refer to food experiences in the last three months. The household undernourishment indicator uses external definitions of adequate energy needs, whereas ELCSA involves assessing food access based on households’ self-reported experiences, perceptions, and awareness. Relative to the ELCSA, the amount of information needed for the household undernourishment indicator is very large and the procedure involved for its calculation is significantly more complex. Despite these differences, both indicators provide estimates of food insecurity prevalence during the year the HES was implemented. 

Each food security dimension (in this case, access) is composed of several sub-dimensions. Sub-dimensions of the access dimension of food security identified in the literature include food quantity, food quality, households’ anxiety about food access, households’ ability to cope with food shortfalls, and cultural acceptability [2]. In our case, the household undernourishment indicator measures net energy availability (which depends on food quantity and quality), whereas ELCSA reflects households’ perceptions about food quality and quantity, their anxiety about food access, and coping strategies (see Appendix A). 

Finally, it is important to point out that given the advantages and disadvantages of the indicators, neither ELCSA nor the household undernourishment indicator can be considered a definite or “gold” standard measure for the access dimension of food security. Therefore, our analyses compare the relative empirical performance of the indicators against each other.

## 2. Data and Methods 

Data for our analyses come from the 2011 National Survey of Living Standards (ENCOVI-2011) from Guatemala. The survey was conducted between March and August, reached 13,531 households, and used stratified random sampling. ENCOVI’s main objective is to estimate the prevalence of poverty in the country; thus, the ENCOVI-2011 survey sampling design was based on the variable measuring households’ extreme poverty status and aimed at estimating its prevalence at the departmental and regional levels with a 95% level of confidence [19].

ENCOVI collects information about a variety of topics related to households’ characteristics and living conditions. The survey is divided into 15 chapters. Information for this study was collected from the survey introduction (household and geographic identifiers), chapter I (households’ dwellings and conditions), chapter IV (household members’ characteristics), and chapter XII (expenditures and self-consumption). Unless otherwise specified, all variables were used as given in the survey. 

ELCSA questions were included at the end of chapter I (see Appendix A). Data for the construction of undernourishment was obtained from chapter XII of the survey, which collects data on households’ acquisitions (quantities purchased on the market or obtained from other sources and corresponding expenditures) of 116 food products in the previous fifteen days. Chapter XII also collects data on total households’ expenditures during the last week on 30 goods/services frequently consumed (e.g., public transportation), and annual household expenditures on 31 categories of durable or nondurable goods purchased less frequently (e.g., apparel and home repairs). Information on expenditures on all spending categories is used by Guatemala’s National Institute of Statistics (INE) to calculate annual household expenditures and categorize a household poverty status. Categorization of households as poor uses per-capita poverty line cut-offs, which are pre-defined minimum levels of expenditures needed to purchase some basic necessities of life. Following the standards used by the Guatemalan government, the poverty line on a per-capita basis in 2011 was set at $1192/year while the extreme poverty line was set at $578/year. By this measure, in 2011, more than half the population in Guatemala lived in poverty conditions, with 41% of the population living in poverty and 13% in extreme poverty [19]. 

The nutrition information on the food items included in ENCOVI (needed to calculate the household undernourishment indicator) was obtained from the Table of Nutritional Composition of Central American Foods [20]. The table provides information on 28 nutrients for 1169 food products in the region. Additionally, information on human energy requirements was collected to estimate the household’s energy requirements [21]. 

### 2.1. Calculation of the Indicators 

The procedures for calculating ELCSA and the household undernourishment indicator are substantially different, yet both indicators result in households’ classification as either food secure or insecure. As can be seen from Figure 1, the estimation procedure of the food security status of a household is more straightforward using ELCSA than with IFPRI’s method.

In the household undernourishment case, we followed the method suggested by Smith and Subandoro, where a household is classified as food secure if its energy balance is positive or zero and as food insecure if its energy balance is negative. In the case of ELCSA, households that respond negatively to all questions in the instrument are classified as food secure (see Table 1). If they respond affirmatively to at least one of the questions, they are classified as food insecure [4].

### 2.2. Logistic Regression

To evaluate the effect of socio-demographic characteristics on household food security status, we estimated logistic regression models using ELCSA and IFPRI’s undernourishment indicators as dependent variables. For each dependent variable, we estimated two regression models, one for the entire population and another only for poor households since they are the focus of our policy simulations. The probability of households being food insecure can be expressed as π=Pr(Y=1|x), where ***x*** is a vector of explanatory variables. The specific functional form for the probability Pr is given by (Greene 2012):(1)$$\mathrm{Pr}(Y=1|x)=\frac{\exp\left(\alpha +\beta^{\prime }x\right)}{1+\exp\left(\alpha +\beta^{\prime }x\right)},$$
where α and ***β*** are parameters to be estimated. Model parameters are estimated using maximum likelihood methods [22]. The selection of explanatory variables was based on the literature review and objectives of the study. Thus, the vector ***x*** included characteristics of the household and household head, household location, total annual expenditures, and poverty status (see Table 2) [3,6]. In addition to these variables, we also included a variable related to the timing of the survey. More specifically, we constructed a dummy variable indicating when the survey was taken during the lean season (July–August), which is the period of the year in which poor rural households are most food insecure and most likely to require food aid [23]. Additionally, to add flexibility to the model, we added an interaction between the survey timing variable and a dummy variable indicating a rural location of a household, and an interaction between household poverty status and total income. These interactions allow us to account for more flexible marginal effects of the variables included in the interactions.

The coefficients in the logistic regression model have interpretations in terms of the odds (or log-odds) of households’ being food insecure since $log\left(\frac{\pi }{1-\pi }\right)=\alpha +\beta^{\prime }x$; however, a more intuitive interpretation of logistic model results can be obtained using marginal effects (i.e., the derivatives) which measure the effect of one unit change in a variable on the probability of being food insecure. Formally, the marginal effect of a variable ***x_j_*** is calculated as follows:(2)$$\frac{\partial \mathrm{Pr}(Y=1|x)}{\partial x_{j}}=\frac{\exp\left(\alpha +\beta^{\prime }x\right)}{1+\exp\left(\alpha +\beta^{\prime }x\right)}\beta_{j}$$
where $\beta_{j}$ is the term corresponding to the ***x_j_*** explanatory variable. Marginal effects for binary variables are formally obtained as changes in the probabilities’ values when a variable change from 0 to 1; however, the formula in Equation (2) provides a very accurate approximation of these marginal effects and is used in practice [22]. The marginal effect of a variable $x_{i}$ interacted with a variable $x_{k}$ is given by
(3)$$\frac{\partial \mathrm{Pr}(Y=1|x)}{\partial x_{i}}=\frac{\exp\left(\alpha +\beta^{\prime }x\right)}{1+\exp\left(\alpha +\beta^{\prime }x\right)}(\beta_{i}+\beta_{ik}x_{k})$$
where $\beta_{i}$ is the term corresponding to the *x_i_* explanatory variable and $\beta_{ik}$ is the coefficient related to the interaction between *x_i_* and $x_{k};$ thus, in the model with interactions the effect of $x_{k}$ on the marginal effect of *x_i_* is more flexible (i.e., it can have the opposite sign that the $\beta_{h}$ coefficient). Marginal effects were evaluated at every observation. The sample average and standard errors of the individual marginal effects are reported. The SAS^®^ 9.4 software LOGISTIC, QLIM, and MEANS procedures were used to estimate model coefficients, marginal effects, and summary statistics.

### 2.3. Policy Simulations

The policy simulations evaluate the change in the probability that poor Guatemalan households are food insecure as a result of a hypothetical government cash transfer, and thus it uses the results of the regression analyses. Our cash transfer amount was set at $25/month (~Q. 188) per household, independently of the number of household members. We chose this amount based on the value of observed conditional cash transfers in the region [24]. The cost of implementing a policy of this magnitude in Guatemala adds up to $39 million per month, excluding administrative costs. The 2019 budget from the Guatemalan Ministry of Public Health and Social Assistance included about 135 million dollars to prevent infant mortality and nutrition. The total budget for the Ministry is about $1.1 billion. Therefore, the assumed cash transfer program is very large relative to current health and social programs’ expenditures. On the other hand, $25/month is only about 7% of the country’s monthly minimum wage [25].

Before simulating the effect of the cash transfer on food insecurity, the models’ in-sample predictive power was evaluated using sensitivity and specificity measures. Sensitivity is the proportion of correctly predicted events, in this case, food-insecure households, and specificity is the proportion of non-events that are correctly predicted, in our case, the food secure households. Following Allison’s recommendation, the probability cut-off points (i.e., the predicted probability values used to classify the household as food insecure/secure) were adjusted so that the models would provide similar levels of sensitivity and specificity [26]. Cut-off points of 0.5 resulted in 0% sensitivity and 100% specificity in ELCSA and 19% sensitivity and 95% specificity in household undernourishment. Therefore, the used cut-off points were 0.1 for ELCSA and 0.24 for household undernourishment. In the end, the sensitivity and specificity of the ELCSA model were 60% and 62%, respectively, and for household undernourishment were 67% for both measures.

The simulated effects of the cash transfers were estimated as follows. First, for each household member, we calculated the predicted probability $\hat{\mathrm{Pr}}$ of being food insecure using the estimated model coefficients $\hat{\alpha }\;\mathrm{and}\;\hat{\beta }$ and the original values of the explanatory variables (***x***). Using an upper index to identify a specific household ***h***, the predicted probability of being food insecure is
(4)$$\hat{\mathrm{Pr}}\left(Y^{h}=1|x^{h}\right)=\frac{\exp\left(\hat{\alpha }+{\hat{\beta }}^{\prime }x^{h}\right)}{1+\exp\left(\hat{\alpha }+{\hat{\beta }}^{\prime }x^{h}\right)},\;$$
where $x^{h}$ represents the original values of the explanatory variables for the *hth* household. To calculate the predicted value under the assumed cash transfer, the total expenditure variable in $x^{h}$ was modified to include the cash transfer value. Denoting the vector of household characteristics with the modified total expenditures value as $x_{modified}^{h}$, the probability of being food insecure is then
(5)$$\hat{\mathrm{Pr}}\left(Y^{h}=1|x_{modified}^{h}\right)=\frac{\exp\left(\hat{\alpha }+{\hat{\beta }}^{\prime }x_{modified}^{h}\right)}{1+\exp\left(\hat{\alpha }+{\hat{\beta }}^{\prime }x_{modified}^{h}\right)}$$

Predicted probabilities using Equations (4) and (5) and the previously discussed cut-off points were subsequently used to calculate predicted values for food insecurity prevalence with and without the cash transfer for each of the country’s eight development regions. The difference in the predicted insecurity prevalence levels can be interpreted as the simulated effect of the policy on food insecurity prevalence.

## 3. Results and Discussion

At the national level, food insecurity prevalence was estimated at 83.3% and 61% using ELCSA and household undernourishment indicators, respectively. When analyzing food insecurity prevalence by region (see Figure 2), it can be observed that ELCSA consistently yields higher estimates of food insecurity across all regions. ELCSA food insecurity prevalence estimates are, on average, 22.1% higher than prevalence estimates using the household undernourishment indicator. The smallest difference between prevalence estimates is the Metropolitan areas (7.1% difference) area and the largest in the Northwest (44.3% difference).

When multiplying the population of the regions by the prevalence estimates of food insecurity, it is estimated that 12,376,496 and 9,051,756 people are food insecure, according to ELCSA and household undernourishment, respectively. Thus, more than 3 million people may or may not be categorized as food insecure, depending on the indicator used.

The indicators also identify different regions as being the most food insecure. Regional prevalence estimates using ELCSA identify the Northwest, Petén, Southeast, and Central regions as the top four most food-insecure regions. According to the household undernourishment indicator, the top four most food-insecure regions are the North, Petén, Northeast, and Metropolitan regions. The only region identified by both indicators as highly food insecure was Petén. On the other hand, ELCSA identified the Northwest as the region with the highest prevalence of food insecurity, whereas according to the household undernourishment indicator, this is the region with the lowest prevalence of food insecurity. These observed differences could lead to differences in selecting priority regions to implement programs or projects addressing food insecurity.

Regarding individual households’ food security classification, both indicators only agree on 59.2% of households’ food insecurity status, whereas the remaining 40.8% of households are classified differently (Table 3). This finding is consistent with Jimenez et al. ’s study in Colombia, where the ELCSA only correctly predicted between 46 and 62% of undernourished households [11].

Additionally, to the estimation of the prevalence of food insecurity by region, we also estimated the prevalence of food insecurity by income groups (Figure 3). The food insecurity prevalence estimate using the household undernourishment indicator is consistently lower than the estimate using ELCSA’s across all income groups. However, both indicators resulted in high food insecurity prevalence estimates across all income groups (above 30%). For example, according to ELCSA, 64% of households in the fifth income quintile are food insecure. This seems to support Maxwell et al. conclusion that ELCSA captures psychological anxiety and preferences, which are part of being food insecure but are not severe manifestations of food insecurity, and therefore result in higher estimates of the food insecurity prevalence when compared to undernourishment, even for the highest income groups [5]. In the household undernourishment indicator, the high prevalence of food insecurity in the higher income groups is attributed to under-reporting of food expenditures during the data collection process, leading to underestimating energy availability [4].

### 3.1. Logistic Regression Models

We found that the most likely profile of food-insecure households were different depending on the model used, with disagreement in variables such as gender of the head of the household, number of household members and area (rural–urban). We first present and contrast the two logistic regression results, modeling the probability that a household is food insecure using ELCSA and the household undernourishment indicators and using data representative of the entire country’s population (Table 4). We present both the model parameter estimates as well as marginal effects. In the logistic regression model, the parameter estimates corresponding to continuous variables are interpreted as the change in the log-odds of being food insecure for every one-unit increase in the value of the explanatory variable, ceteris paribus. Parameters corresponding to dummy variables are interpreted as differences in the log-odds of being food insecure between households for which the dummy variable is one and the baseline category. For example, in our models, the excluded regional variable is Metropolitan. Thus, the North regional variable in the ELCSA model indicates that the log-odds are 0.6518 lower for households located in the North region relative to households located in the Metropolitan region. Alternatively, the average marginal effects measure changes in the probability of being food insecure. For example, the interpretation of the parameter estimate of a continuous variable, such as the number of household members in the household undernourishment model, is that for every additional household member, the probability of the household being food insecure increases by 5%. The interpretation of the average marginal effect of a dummy variable such as poverty is that, on average, the probability that poor households are food insecure is 47% higher with respect to non-poor households, ceteris paribus.

When comparing the regression models, it is important to highlight that, in most cases, the sign of the estimated coefficients differs across models. Out of 19 estimated coefficients, only 3 have the same sign in both models: coefficients corresponding to the dummy variable identifying poverty status, per-capita annual expenditures and the interaction of both variables (Table 4). The coefficients related to the variables North, number of household members, educational variables, and the interaction between rural and July–August were negative in the ELCSA model and positive in the household undernourishment model. The positive effect of education on food security using the household undernourishment indicators was an unexpected result. We conjecture that this result may be due to the method used for the construction of this indicator that is based on two elements: household energy requirements and household energy availability (energy availability—energy requirements). Each of the factors included in the regression analysis, including education, affects both elements; thus, the final estimated effect on the probability of undernourishment reflects the net effect of a factor on energy availability and energy requirements. More work is needed to better understand the source of this result (e.g., see [27] for a potential approach for the analyses). The opposite happened with the coefficients related to the variables Northeast, Southeast, Central, Southwest, Northwest, Petén, female, indigenous, rural, July–August, and children’s presence, which were found to be positive in the ELCSA model and negative in the household undernourishment model. Regarding the statistical significance of the variables, while most of the variables were significant in both models, the number of household members and the interaction between rural and July–August were statistically significant in the household undernourishment model but not in the ELCSA’s model. The opposite happened for the variable female.

Finally, we also found relatively large differences in the magnitude of the effects for variables whose coefficients had the same sign in both models (poverty status and per-capita annual expenditures). Both parameter estimates and marginal effects for both variables are higher, in absolute values, in the household undernourishment indicator model, suggesting they play a more significant role in the probability of the households being food insecure when using the household undernourishment model than when using the ELCSA’s model.

Because of their similitude with the general models, the logistic regression models’ results only for the poor are not shown. However, as in the case of the models estimated for the entire population, the signs of most of the estimated coefficients differ across models (14 out of 18 coefficients). The variable female (with a positive effect) had the same sign in both models, which was different from that found in the general population models, yet it was not statistically significant in the household undernourishment model. All other variables exhibited the same pattern as in the general models.

In short, according to the models estimated using ELCSA, poor households whose head is female and/or indigenous with a low educational level, located in the rural area, with children, small household size and with low income are more likely to be food insecure, especially during the lean season. On the other hand, the models estimated using the household undernourishment indicator find that large households with low income and located in the urban area are more likely to be food insecure.

The difference in the estimation results can be explained by the difference in the specific sub-dimension of food access measured by both indicators. Whereas ELCSA is an experienced-based food insecurity indicator, the household undernourishment indicator refers to calorie availability and requirements. For this perspective, for example, the ELCSA model indicates that households experience more food insecurity outside the Metropolitan area of Guatemala, but they are not necessarily more likely to be calorie deficient according to the household undernourishment indicator. Similarly, although indigenous and rural households experience more food insecurity, they are between 7 and 11% less likely to be calorie deficient according to the undernourishment indicator.

### 3.2. Policy Simulation

Simulation results show that the assumed cash transfer to poor households in Guatemala results in a total reduction of 2.4% and 7.5% in food insecurity levels when using the models estimated using ELCSA’s and the household undernourishment indicator-based models, respectively (Figure 4). The simulation results also show consistently higher reductions in food insecurity as a result of the cash transfer program when using the household undernourishment model across all the regions in the country.

Despite yielding different quantitative results, both simulation analyses suggest relatively small reductions in the food insecurity prevalence as a result of the assumed cash transfer, which can be explained by the relatively small magnitude of the per-capital annual expenditure coefficients in the logistic regression models. Thus, only very large cash transfer programs can result in high levels of reduction in food insecurity prevalence.

## 4. Summary and Conclusions

This study’s primary objectives were to empirically compare two alternative food security indicators (ELCSA and household undernourishment) to measure the prevalence of food insecurity, assess factors associated with household food insecurity, and evaluate (ex ante) the potential impact of a cash transfer policy. Data for the study comes from a nationally representative survey of Guatemalan households conducted in 2011 that collected information to calculate the two food security indicators.

Our results show that even though both indicators operate in the same dimension of the concept of food security (access) and at the same level (households), they yield different estimates of the prevalence of food insecurity and also significantly differ (in 40% of cases) in the households’ food insecurity status. This disagreement results in differences in the estimates of food insecurity prevalence across regions and income levels. Logistic regression models estimated to assess and identify household food insecurity drivers also found large differences in both the direction and magnitude of factors affecting food insecurity using the alternative food security indicators. Finally, policy simulation results show similar qualitative results when using both indicators: the cash transfer policies are likely to have only a small effect on reducing food insecurity. However, the magnitude of the impact was significantly larger when using the undernourishment indicator. This also suggests that increasing income alone, via cash transfers, may not be an efficient policy for improving food security and that a holistic approach is necessary if Guatemala is to take important steps towards reducing the incidence of food insecurity. This holds true independently of the methodology used to measure food security.

Although the differences found in the analyses’ results using both indicators are very troubling at first sight, we believe it reflects substantial dissimilarities in their theoretical foundation and method. Therefore, these results should be taken as a cautionary message for researchers and policymakers against the use of these two food security indicators, or any other food security indicators, interchangeable, and without a proper understanding of their theoretical foundation and method, even when the indicators operate in the same food security dimension. It is also imperative for researchers to be exact when referring to food security indicators. Moreover, the dimension and sub-dimension of food security being measured should always be made very clear.

Regarding the empirical results of the study as they relate to Guatemala, we consistently find evidence of very high levels of food insecurity in the country, which is consistent with assessments obtained by other institutions using alternative indicators of food security [28]. Although both ELCSA and the household undernourishment indicator have been validated in the past, some of the empirical findings are problematic and difficult to explain. We found high estimates of food insecurity across income groups with ELCSA, resulting in what seems to be even implausible levels of prevalence of food insecurity in the fifth quintile of the income distribution. Therefore, we believe it would be premature to use the study results for policy design without a proper understanding of the indicators. This study’s results can be cross-checked using the most recent wave of ENCOVI, which was conducted in 2014. This also highlights the need for more research to further evaluate the performance of ELCSA and the undernourishment indicator using large nationally representative surveys. Whereas some body of literature has evaluated the use of HES to measure nutritional outcomes [4,29,30], most of the literature evaluating ELCSA focuses only on its internal reliability; thus, more work is needed to evaluate this indicator [2,8,9].

Future work comparing ELCSA and the undernourishment indicator can also be conducted using alternative cut off points for the classification of food-insecure households. For example, using ELCSA’s severe food insecurity category to classify households as food insecure (instead of all three ELCSA’s food insecurity categories) is more likely to involve actual food deprivation, and thus be more closely related to the household undernourishment indicator. More generally, additional research is needed to compare alternative food security indicators for purposes beyond the estimation of food security prevalence; however, it is important to emphasize that this type of comparison requires data collection for the calculation of indicators from the same household. The comparison of the indicators could also be conducted using longitudinal data instead of cross-sectional data in order to evaluate the evolution of the indicators within households across time. More research is also needed to better understand the sub-dimension components of currently used food security indicators and their contribution to the overall food security indicator. For example, when using the undernourishment indicator as a measure of the food access dimension of food security, it would be interesting to disentangle the separate effects of the amount of food being consumed (quantity) and the quality of food. Similar work can be conducted with ELCSA to separate the food quantity and quality sub-dimensions from the sub-dimensions related to anxiety and sensations about food access. Although several arguments can be made in favor or against using one indicator or the other, including aspects related to implementation costs or reliability, the choice of one indicator over another should ultimately be based on policy objectives. Several alternative food security indicators within each food security dimension should be used for policy analysis and implementation when possible. Ideally, such indicators would complement each other and provide a whole picture of the food security dimension targeted by the policy. Finally, it is important to emphasize the fact that food security is a multi-dimensional concept. A complete understanding of the food security problem requires the study of all the dimensions and sub-dimensions of the problem and the use of multiple indicators.

## Figures and Tables

**Figure 1 nutrients-12-03307-f001:**
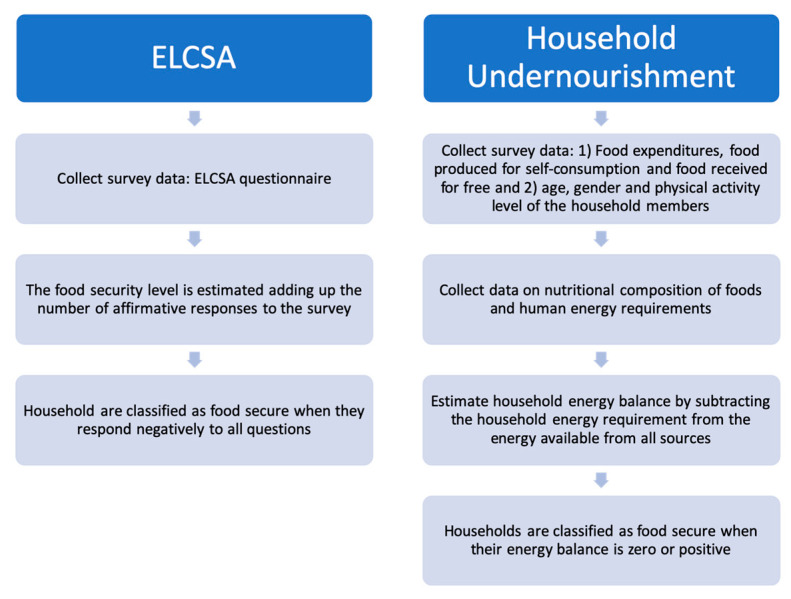
Estimation procedure of ELCSA and household undernourishment food security indicators. ELCSA, Latin American and Caribbean Food Security Scale.

**Figure 2 nutrients-12-03307-f002:**
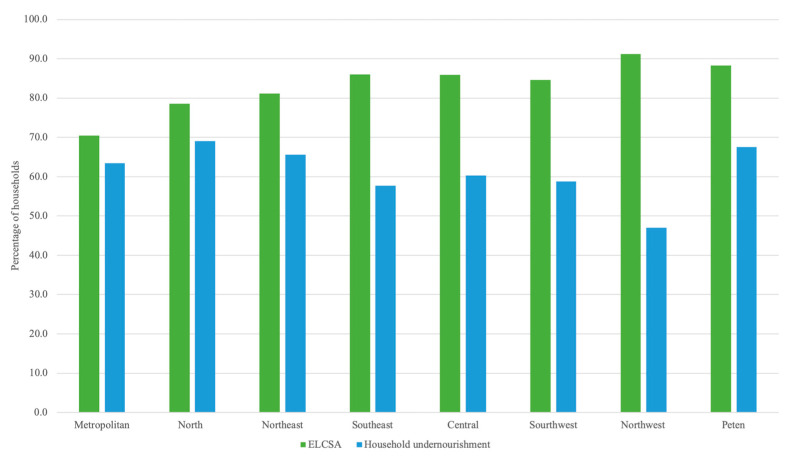
Food insecurity prevalence by region. ELCSA, Latin American and Caribbean Food Security Scale.

**Figure 3 nutrients-12-03307-f003:**
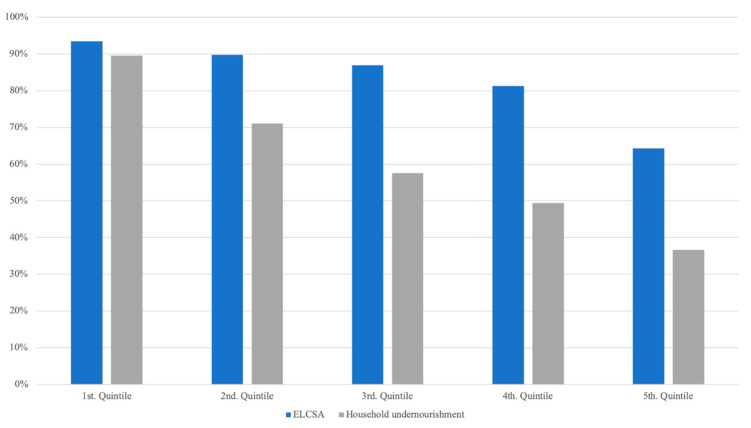
Prevalence of food insecurity across income groups. ELCSA, Latin American and Caribbean Food Security Scale.

**Figure 4 nutrients-12-03307-f004:**
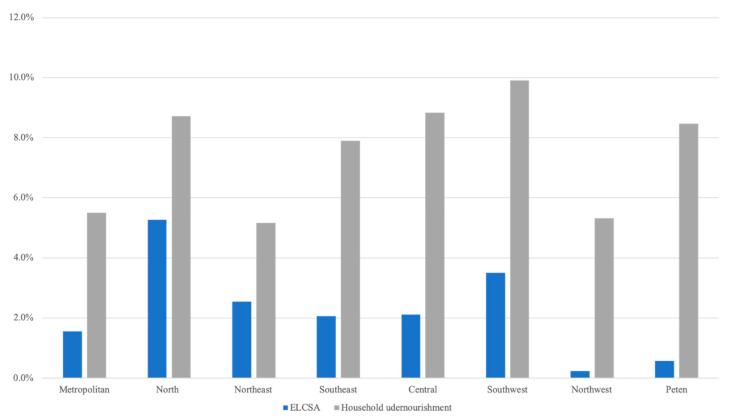
Percentage reduction in the incidence of food insecurity from a simulated cash transfer policy. ELCSA, Latin American and Caribbean Food Security Scale.

**Table 1 nutrients-12-03307-t001:** Latin American and Caribbean Food Security Scale (ELCSA)’s food security categories and required affirmative answers. Source: [6].

Category	Households with Children	Households without Children
Food secure	0	0
Mild food insecure	1–5	1–3
Moderate food insecure	6–10	4–6
Severe food insecure	11–15	7–8

**Table 2 nutrients-12-03307-t002:** Explanatory variables for the logistic regression models.

Category	Variable	Mean	Standard Deviation	Minimum	Maximum
Continuous variables	Number of household members	4.934	2.451	1.000	22.000
	Per-capita annual expenditures (Thousands of Q *)	11.797	12.356	0.535	305.999
Dummy variables	North region	0.069	0.254	0.000	1.000
	Northeast region	0.224	0.417	0.000	1.000
	Southeast region	0.103	0.304	0.000	1.000
	Central region	0.157	0.364	0.000	1.000
	Southwest region	0.271	0.444	0.000	1.000
	Northwest region	0.071	0.256	0.000	1.000
	Petén region	0.031	0.173	0.000	1.000
	Female	0.206	0.405	0.000	1.000
	Indigenous	0.344	0.475	0.000	1.000
	Rural area	0.586	0.493	0.000	1.000
	July–August (lean season)	0.200	0.400	0.000	1.000
	Presence of Children (under 18 years)	0.792	0.406	0.000	1.000
	Primary and middle school education	0.651	0.477	0.000	1.000
	University education	0.035	0.183	0.000	1.000
	Poor household (as measured by the government)	0.54	0.498	0.000	1.000

* Exchange rate as of 30 June 2017: US$ 1 = Q. 7.34.

**Table 3 nutrients-12-03307-t003:** Classification of households by ELCSA and household undernourishment.

		ELCSA	
		Food Secure	Food Insecure
Household undernourishment	Food secure	7.5%	31.6%
	Food insecure	9.2%	51.7%

**Table 4 nutrients-12-03307-t004:** Parameter estimates of logistic models for the food insecurity status of households using data representative of the entire population.

Variable	Parameter Estimates		Average Marginal Effects	
	ELCSA	Undernourishment	ELCSA	Undernourishment
Intercept	1.1895 *** (0.1295)	0.4481 ** (0.1244)		
North	−0.6518 *** (0.1222)	−0.3423 ** (0.1140)	−0.0806 (0.0386)	−0.0631 (0.0222)
Northeast	0.0694 (0.0933)	−0.2403 ** (0.0869)	0.0086 (0.0041)	−0.0443 (0.0156)
Southeast	0.3005 ** (0.1138)	−0.8418 *** (0.0990)	0.0372 (0.0178)	−0.1551 (0.0547)
Central	0.4885 *** (0.1005)	−0.5215 *** (0.0888)	0.0604 (0.0290)	−0.0961 (0.0339)
Southwest	0.0007 (0.0939)	−0.8206 *** (0.0864)	0.0001 (0.00004)	−0.1512 (0.0533)
Northwest	0.4840 ** (0.1457)	−1.3942 *** (0.1120)	0.0599 (0.0287)	−0.2569 (0.0906)
Petén	0.4154 ** (0.1787)	−0.3370 ** (0.1409)	0.0514 (0.0246)	−0.0620 (0.0219)
Female	0.1885 ** (0.0630)	−0.0395 (0.0506)	0.0233 (0.0112)	−0.0073 (0.0026)
Indigenous	0.3394 *** (0.0652)	−0.4398 *** (0.0515)	0.0420 (0.0201)	−0.0811 (0.0286)
Rural	0.4302 *** (0.0641)	−0.4888 *** (0.0566)	0.0532 (0.0255)	−0.0901 (0.0318)
July–August	0.2824 ** (0.0922)	−0.1662 ** (0.0769)	0.0349 (0.0167)	−0.0306 (0.0108)
Rural*July–August	−0.1329 (0.1296)	0.2992 ** (0.1022)	−0.0164 (0.0079)	0.0551 (0.0194)
Presence of Children	0.4063 *** (0.0655)	−0.4940 *** (0.0576)	0.0502 (0.0241)	−0.0910 (0.0321)
Primary and middle school education	−0.3695 *** (0.0633)	0.2073 *** (0.0481)	−0.0457 (0.0219)	0.0382 (0.0135)
University education	−1.0770 *** (0.1235)	0.5842 *** (0.1223)	−0.1332 (0.0639)	0.1077 (0.0380)
Number of household members	−0.0188 (0.0139)	0.2718 *** (0.0124)	−0.0023 (0.0011)	0.0501 (0.0177)
Poverty	1.1116 *** (0.1360)	2.5525 *** (0.1095)	0.1375 (0.0659)	0.4704 (0.1659)
Per-capita annual expenditures	−0.0222 *** (0.0025)	−0.0413 *** (0.0032)	−0.0027 (0.0013)	−0.0076 (0.0027)
Poverty * Per-capital annual expenditures	−0.0686 *** (0.0166)	−0.2390 *** (0.0137)	−0.0085 (0.0041)	−0.0440 (0.0155)

Standard Errors shown in parenthesis. *, **, ***, denote significance at 0.1, 0.05, and 0.0001, respectively.

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
