# Peer review of "Comparison between Experience-Based and Household-Undernourishment Food Security Indicators: A Cautionary Tale"

_nutrients, 2020, doi:10.3390/nu12113307_

Round 1

Reviewer 1 Report

I thought that the article did a good job addressing the questions involved with indicators. Hopefully other reviewers can address the questions of content and methodology.  Here are a few additional comments and I hope that they are helpful:   

This article examines the reliability of indicators in measuring real world phenomena. In this case, the authors look at two indicators that assess the level of food insecurity in Guatemala. The authors show that the two different indicators actually produce different results even though they are measuring the same thing.

This finding has important implications for policy makers. How can they make decisions based on evidence if the evidence that they receive is not trustworthy? Or if the results obtained really depend on what kind of instrument is being used? The paper is important in demonstrating that we cannot rely heavily on indicators unless we have a strong idea of what is really behind them.   

This article is important because it is a contribution to a debate about the importance of indicators and quantitative data in general. More and more, policy makers are trying to use objective information in making decisions. Accordingly, indicators in general have become very popular and are found throughout our lives.

This paper is part of a growing body of literature that shows the limitations of the indicators that we use and the kinds of cautions that we should put in place when we use them. The ideas are not particularly original, but they help add nuance to our understanding of indicators and their practical application.   

The conclusions are consistent with the data presented and provide guidance for thinking about how to use these particular indicators. While this article has a very specific focus, it may be helpful for other scientists who are interested in indicators because it provides another data point in understanding how the indicators are used.   

Overall, this is an interesting and well written article. I think that its point about how different indicators produce different results even when measuring the same phenomenon is valid and well argued.

Author Response

Reponse to reviewer #1

We would like to thank reviewer #1 for his/her valuable comment. 

Reviewer #1 only made comments about the manuscript and no specific changes were requested. However, he/she indicated that minor spell check is required. Therefore, we have double checked the manuscript. All changes resulting changes from the spell check are shown in the manuscript.

Reviewer 2 Report

Overall this paper is clearly written and appears to fill a gap in the research about the use of food security indicators, including a policy example.

Abstract

Write ELCSA in full the first time it appears

Is the food security indicator calculated using household expenditure surveys or available from household expenditure surveys? This sentence is not clear.

Indicate in the abstract the direction of results from the indicators – did one over or underestimate food security?

Introduction

Second paragraph: Replace ‘their’ in the first sentence with household to be clear that it refers to the household, not the adult answering the questions.

Literature review – While much of the information is important it is too long so loses the reader's attention, and some information, such as the details of the studies could be in the discussion. I suggest the literature review is reviewed and reduced in length. 

1.3: IFPRI in full

Table 2 – the description column seems unnecessarily wordy as the variables seem reasonably self-explanatory, consider if there is a way to shorten the description (or remove this column all together).

2.3 Reference Allison earlier in the sentence

Table 4: What does - in the table mean? It is difficult to work out which row of figures belongs to which variable. I suggest adding a slight gap between rows belonging to different variables so this is clear. Check that the data is in the correct row as not all data seems to line up.

3.1: The last 2 paragraphs important as they describe the key results and explanation but because these are at the end of the section they are somewhat lost. I suggest having a summary statement at the beginning of 3.1 to describe the results.

Summary: The authors briefly mention in the final paragraph that several food security indicators could be used. Please discuss this further considering the two indicators used in this study. Do you recommend the indicators be used together? Can they complement each other since they measure different aspects?

Who is the reviewer in the last paragraph of the summary that made suggestions for use of ELCSAs co-categories?

References: Ref 14 is in capitals, change to title case.

Author Response

Response reviewer #2

We would like to thank reviewer #2 for his/her valuable comments and recommendations. In additions to the comments and recommendations, reviewer #2 indicated that minor spell checking is required and that the presentation of the results can be improved. We present our responses in cursive.

Abstract

Write ELCSA in full the first time it appears

The food security indicator ELCSA was written in full the first time it appears in the abstract. Next to it, we added (ELCSA, by its acronym in spanish).

Is the food security indicator calculated using household expenditure surveys or available from household expenditure surveys? This sentence is not clear.

You are right. The sentence you refer to is confusing. We deleted it an added relevant information in the following sentence. We added: “…which collected all the data for estimating the ELCSA and the household level data required for estimating the household undernourishment indicator.”

Indicate in the abstract the direction of results from the indicators – did one over or underestimate food security?

 We added “…with ELCSA resulting in higher estimates” to the sentence where we discuss the estimated prevalence of food insecurity.

Introduction

Second paragraph: Replace ‘their’ in the first sentence with household to be clear that it refers to the household, not the adult answering the questions.

We replaced “their” with “the household´s”.

Literature review – While much of the information is important it is too long so loses the reader's attention, and some information, such as the details of the studies could be in the discussion. I suggest the literature review is reviewed and reduced in length. 

We circulated early versions of the paper among colleges for feedback. We found that a shorter literature review resulted in the readers asking us about previous food security indicators comparisons and what our contribution was to those previous comparisons. While we agree with you that it could be shorter, the detailed explanation is actually necessary for some of the readers not familiar with this literature. We would like to keep the current length of the literature review.

With respect to your comment that such details of the studies could be in the discussion, we have actually recalled [4, 5, 11] when interpreting our results.

1.3: IFPRI in full

We added: International Food Policy Research Institute (IFPRI).

Table 2 – the description column seems unnecessarily wordy as the variables seem reasonably self-explanatory, consider if there is a way to shorten the description (or remove this column all together).

We deleted the description column and expanded the names of the variables to help the reader understand them.

2.3 Reference Allison earlier in the sentence

We referenced Allison in the same sentence is it mentioned instead of the end of the paragraph.

Table 4: What does - in the table mean? It is difficult to work out which row of figures belongs to which variable. I suggest adding a slight gap between rows belonging to different variables so this is clear. Check that the data is in the correct row as not all data seems to line up.

As you, we got confused with the “-“ symbol. It actually is a negative sign that did not fit in the column. We adjusted the width of the columns so that the whole number would fit, including the sign, four decimals and significance level. We will consult the editor with your request of adding a slight gap between rows, as it seems and editorial decision.

3.1: The last 2 paragraphs important as they describe the key results and explanation but because these are at the end of the section they are somewhat lost. I suggest having a summary statement at the beginning of 3.1 to describe the results.

We added the following statement at the very beginning of section 3.1: “We found that the profile of the most likely to be food insecure households was different depending on the model used, with disagreement in variables such as gender of the head of the household, number of household members and area (rural-urban), among others.” We believe this statement summarizes the results without giving them away and without being repetitive of the last two paragraphs.

Summary: The authors briefly mention in the final paragraph that several food security indicators could be used. Please discuss this further considering the two indicators used in this study. Do you recommend the indicators be used together? Can they complement each other since they measure different aspects?

To further discuss our recommendation of using several food security indicators, we added the sentence: “Ideally, such indicators would complement each other and provide a whole picture of the food security dimension targeted by the policy.”

Who is the reviewer in the last paragraph of the summary that made suggestions for use of ELCSAs co-categories?

We circulated the manuscript among some collaborators before submitting it for peer-review. To prevent this from creating confusion, we changed the sentence to: “For example, using ELCSA´s severe food insecurity category…”

References: Ref 14 is in capitals, change to title case.

We changed the title of the reference to little case.